# Improving the Performance of Index Insurance Using Crop Models and Phenological Monitoring

Mehdi H. Afshar [1,*] , Timothy Foster [1] , Thomas P. Higginbottom [1] , Ben Parkes [1,2] , Koen Hufkens [3] , Sanjay Mansabdar [4] , Francisco Ceballos [5] and Berber Kramer [5]

1 Department of Mechanical, Aerospace and Civil Engineering, University of Manchester, Manchester M13 9PL, UK; timothy.foster@manchester.ac.uk (T.F.); thomas.higginbottom@manchester.ac.uk (T.P.H.); ben.parkes@manchester.ac.uk (B.P.)
2 Centre for Crisis Studies and Mitigation, University of Manchester, Manchester M13 9PL, UK
3 Computational & Applied Vegetation Ecology Lab, Ghent University, 9000 Ghent, Belgium; koen.hufkens@gmail.com
4 Dvara E-Registry, Hyderabad 500003, India; sanjay.mansabdar@dvara.com
5 International Food Policy Research Institute, Washington, DC 20005, USA; f.ceballos@cgiar.org (F.C.); b.kramer@cgiar.org (B.K.)
* Correspondence: mehdi.afshar@manchester.ac.uk

**Abstract:** Extreme weather events cause considerable damage to the livelihoods of smallholder farmers globally. Whilst index insurance can help farmers cope with the financial consequences of extreme weather, a major challenge for index insurance is basis risk, where insurance payouts correlate poorly with actual crop losses. We analyse to what extent the use of crop simulation models and crop phenology monitoring can reduce basis risk in index insurance. Using a biophysical process-based crop model (Agricultural Production System sIMulator (APSIM)) applied for rice producers in Odisha, India, we simulate a synthetic yield dataset to train non-parametric statistical models to predict rice yields as a function of meteorological and phenological conditions. We find that the performance of statistical yield models depends on whether meteorological or phenological conditions are used as predictors and whether one aggregates these predictors by season or crop growth stage. Validating the preferred statistical model with observed yield data, we find that the model explains around 54% of the variance in rice yields at the village cluster (Gram Panchayat) level, outperforming vegetation index-based models that were trained directly on the observed yield data. Our methods and findings can guide efforts to design smart phenology-based index insurance and target yield monitoring resources in smallholder farming environments.

**Keywords:** index insurance; crop yield; APSIM; leaf area index; phenological monitoring

## 1. Introduction

Agriculture plays a critical role in supporting livelihoods and food security for rural households across the developing world [1]. Designing strategies to protect farmers against crop losses caused by adverse weather conditions, such as droughts or floods, has become a key priority for governments and donors, particularly given expected increases in the frequency or intensity of extreme weather events in the coming decades due to climate change [2,3]. One of these strategies is to provide smallholder farmers with agricultural insurance, which offers financial protection from losses associated with extreme weather. In recent years, several agricultural insurance programs have been rolled out at scale, and large amounts of money have been invested in these programs. For instance, in the monsoon season of 2019, India's national insurance scheme, the Pradhan Mantri Fahsal Bima Yojana (PMFBY), covered more than 33.5 million hectares of land through subsidised crop insurance, with gross insurance premiums amounting to more than USD 3 billion.

In developing countries, amongst different types of insurance frameworks (such as whole-farm revenue insurance [4,5] and bancassurance [6,7]), index-based insurance

approaches are the most commonly used within agricultural insurance programmes. Unlike traditional insurance schemes, which are based on the direct verification of crop yield losses for each insured field, payouts from index insurance are made on the basis of an empirical relationship between a proxy index and expected yield losses [8]. Proxies used include weather indices, satellite vegetation indices or area-yield indices, whereby yields are measured for a sub-sample of fields through crop cutting experiments (CCEs) to estimate an average yield over a given region, and payouts are made when these average yields fall below a threshold that is based on historical yields for the region. Using an objective observable index in claims settlement helps provide more timely payouts and reduces costs of loss verification for insurers, making coverage more affordable for farmers and potentially improving farmers' willingness to pay for insurance [9]. Uptake has been generally low, though, in part due to high levels of basis risk, that is, a mismatch between the index—and thus insurance payouts—and actual yield losses [10].

One component of basis risk is design risk, which arises from limited data availability [11]. In particular, the limited availability of observed yield data inhibits the identification and definition of reliable weather and vegetation indices that accurately predict yield losses. Whilst this is not a limitation for area-yield index insurance, high costs of conducting representative samples of CCEs in heterogeneous smallholder farming environments can lead to biased estimates of average yields and thus basis risk. Another important driver of basis risk relates to the temporal specification of the variables used to predict crop yields. Most index-based insurance schemes trigger payouts based on indices that are defined over fixed calendar periods, often relating to the average timing of key phenological stages in a given agricultural system [12–14]. In reality, the timing of a crop's sensitivity to weather may vary significantly across fields due to differences in management practices, such as variety and sowing dates, as well as meteorological conditions, which affect rates of crop development [15]. Failure to consider this heterogeneity may lead to inaccurate estimation of yield losses and basis risk [16,17].

In an effort to address these challenges, we analysed to what extent the integration of crop models and phenological monitoring can help reduce these design and temporal basis risks, respectively. Biophysical crop simulation models can be leveraged to generate larger synthetic yield datasets, which can then be used to train weather- or satellite-based index models [18–20] or support spatial targeting of limited numbers of CCEs that can be conducted as part of area-yield insurance products. However, to date, this approach has not been widely applied in the context of index insurance design, with limited evidence about its performance at spatial scales relevant for insurance applications (e.g., field, farm or village) or in comparison with index models derived empirically from available observational yield datasets. Approaches to reduce temporal basis risks have focused on developed countries, where detailed phenological monitoring networks exist [21]. In contrast, there has been limited attention on how to embed phenological information in the design and implementation of index insurance in smallholder environments, for example, through the use of satellite or in-situ phenology monitoring systems or technologies [22,23].

We addressed these knowledge gaps through a case study on rice yield estimation in the state of Odisha, India, an area of extensive rainfed rice production, where agriculture is highly exposed to risks posed by monsoonal rainfall variability. We demonstrate how the integration of crop models, phenological monitoring through satellite remote sensing, and machine learning techniques can support the design and implementation of smart phenology-based index insurance products at spatial scales relevant for smallholder communities. Our findings highlight the opportunity for robust and scalable yield estimation by combining satellite data with machine learning and crop modelling. We show that this approach can significantly outperform models that rely solely on satellite imagery. At the same time, our results demonstrate several remaining challenges that need to be addressed to accurately and reliably estimate yields at plot scales in smallholder farming environments.

## 2. Materials and Methods

In the following subsections, we outline our methodological approach to estimate rice yields. In Section 2.1, we provide information about the case study area, including key characteristics of agricultural production in Odisha. In Section 2.2, we describe the modelling approach used to develop a database of synthetic yield data for our study area, followed in Section 2.3 by the statistical techniques used to relate simulated yield data to relevant crop, phenology and weather conditions. In Section 2.4, we discuss the process for validating statistical models against both synthetic and observed yield data. We also contrast the performance of our models with estimates of yields derived directly through regression analysis using satellite vegetation indices.

### 2.1. Study Area and Observation Data

Our analysis focused on rice yield estimation in the state of Odisha in eastern India (Figure 1). Agricultural production in Odisha is dominated by small-scale farmers, with most rice production occurring during the summer monsoon season (Kharif). Rice production in the region is mainly rainfed, reflecting the relatively limited access to affordable and reliable irrigation water supplies in eastern India. Monsoonal rainfall variability is, therefore, a key production risk for many farmers. For example, the late onset of the monsoon leads to delays in rice transplanting, resulting in yield losses due to the use of older seedlings and exposure to end-of-season temperature stress [24]. Similarly, a lack of access to irrigation limits farmers' ability to protect crops against dry spells during the season, which can have damaging impacts on yields if droughts occur around critical development stages, such as anthesis and grain filling [25].

To support our analysis of alternative yield estimation approaches, observed yield data were collected through CCEs for a total of 80 paddy rice fields located in two blocks of Jajpur district, Odisha. Yield data were collected in late 2019, following the end of the 2019 summer monsoon season that was characterised by above-average rainfall and early starting time. Rice fields were sampled from 20 village clusters (Gram Panchayats; GPs) as GPs are the primary spatial unit to estimate area-yields in the context of the Indian Government's National Crop Insurance Program (PMFBY). In each of the selected GPs, field staff randomly selected amongst consenting farmers the fields of 5 farmers for seasonal monitoring. Monitoring was done through smartphone images of the crops, and at the end of the season, as the crop had reached maturity, field staff collected yield data through CCEs. Field sizes ranged from 375 to 800 square meters (mean of 630), which are typical of smallholder farming in eastern India.

### 2.2. Synthetic Yield Data Generation

Designing index insurance products requires establishing a relationship between crop yields and one or more predictors (that is, the variables used to operationalise the insurance 'index') that can be observed at a lower cost than would be required to manually verify yields directly through surveys or CCEs. However, the limited availability of in-situ yield data represents a major barrier to estimating these relationships accurately and reliably. Resulting biases in the estimated relationships between yields and predictor variables or indices introduce basis risk. An increase in data availability could help address such basis risk. We, therefore, analysed whether index performance can be enhanced by relying on ensemble process-based crop simulations to generate synthetic yield data, representing yields across a range of potential weather conditions and agricultural management practices that would be infeasible to observe directly through in-situ data collection.

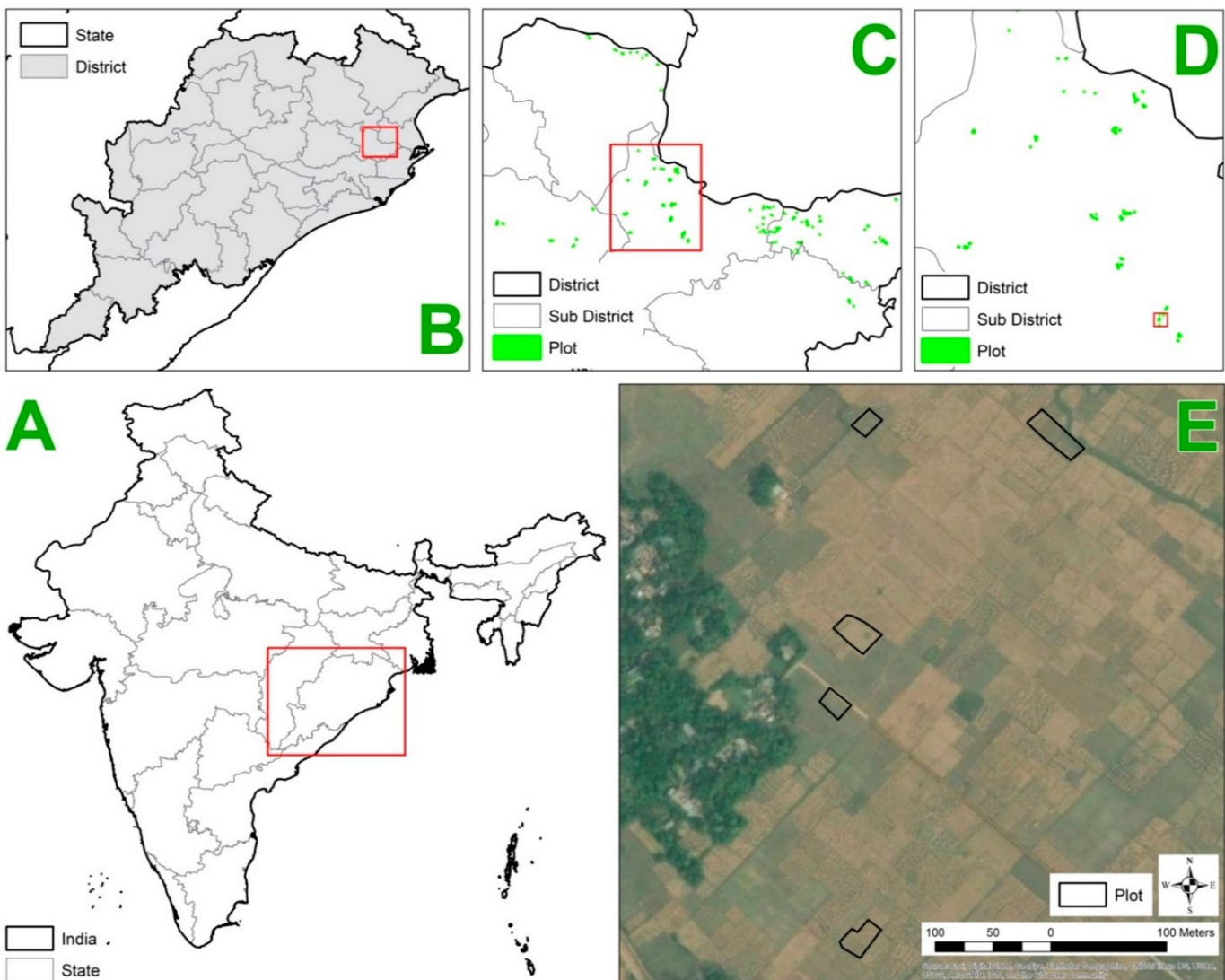

**Figure 1.** The map of the study area with zoom levels at country scale (panel (**A**)), district scale (panel (**B**)), sub-district scale (panels (**C**,**D**)), and plot scale (panel (**E**)).

To develop a database of synthetic yield data, through a process-based crop model—APSIM (Agricultural Production System sIMulator)—we simulated the response of rice yields across a range of potential weather conditions and management practices in our study area. APSIM's rice module, ORYZA2000, is a dynamic physiological model of rice development, which has been widely applied for studies of both rainfed and irrigated rice production systems across south and southeast Asia [26,27]. Thus, the synthetic yield data go beyond the limited observation data stemming from the CCEs, which will be only used to validate the statistical models, not to train the statistical models.

APSIM simulations consider a range of plausible weather and management practice scenarios observed in our study region. Specifically, we varied the parameters in the model specifying sowing dates (from 15 May to 15 August on one-week intervals), seedling ages (from 25 to 40 days on 5-day intervals), planting density (100, 150 or 200 plants per square meter), number of hills (from 30 to 45 hills on 5-hill intervals), and fertiliser amounts (50, 100 or 150 kilograms of urea per hectare) in accordance to information on typical management practices drawn from published literature [24,26] and state-level agronomic advisory documents [28]. We carried out APSIM simulations for each combination of parameter values (2016 in total) for 100 weather years, resulting in a total of 201,600 unique point-based

yield simulations. Weather time series used in the crop simulations were developed using a weather generator (LARS-WG) based on 39 years (1981–2020) of historical observed meteorological data, obtained by averaging time series (including daily minimum and maximum temperature, total precipitation and solar radiation) from ERA5 v5.1.3, at $0.25° \times 0.25°$ resolution, over the 80 plots described in Section 2.1. For each simulation, we defined crop growth parameters in APSIM according to the dominant local rice cultivar—MTU7029. MTU7029 (often referred to as Swarna rice) is a long-duration variety, for which parameters in APSIM have been calibrated and validated previously by Balwinder-Singh et al. (2019) [24]. All simulations assumed that rice was transplanted into a clay loam soil—the dominant soil type for rice production areas in the region based on spatial analysis of soil texture data provided by SoilGrids [29]—with hydraulic properties determined using pedotransfer functions [30]. Specifically, the volumetric water contents used in our analysis for the lower limit, drained upper limit and saturation levels were estimated as 18%, 32.3% and 46.1%, respectively, with saturated hydraulic conductivity, assumed to be 20 mm/day and 1 mm/day for the top five and bottom soil layers (out of six), respectively, in line with APSIM guidelines for ponded transplanted rice simulations [31].

### 2.3. Statistical Yield Models

The variables that are used as predictors of yields in index insurance can vary in terms of the underlying type of data (such as various weather variables or indicators of crop development, such as leaf area index) and the temporal period over which each predictor is aggregated (for instance, whether one uses the average for the entire growing season versus the average for a specific phenological stage), along with the spatial scale at which yields are estimated (plot versus village, GP or district aggregation).

To assess the implications of these choices, we fit 28 alternative statistical models that vary in terms of the underlying assumptions about which variables and what level of temporal aggregation were most useful for explaining variations in the synthetic yield data generated by APSIM, as described in Section 2.2 (Figure 2). Specifically, the 28 alternative model specifications developed consider different unique combinations of three potential meteorological and agronomic predictor variables (that is, temperature: T, precipitation: P, and LAI: L, resulting in seven unique predictor combinations of T, P, L, TP, TL, PL and TPL), along with four different assumptions about the time period over which each predictor was aggregated for yield prediction on a given plot.

The four different temporal specifications that we considered in our analysis include (1) fixed season, FS, representing a typical or 'normal' cropping season between August and November; (2) dynamic season, DS, where variables were aggregated according to plot-specific information about the start and end date of the crop growing season; (3) fixed growth stages, FSTG, where variables were aggregated by fixed crop growth stages timings instead of the timing of the full season; and (4) dynamic growth stages, DSTG, where variables were aggregated by dynamic crop growth stage timings relying on plot-specific information about the start and end dates of each stage. In the FSTG and DSTG models, we divided the rice season into four main stages—transplanting, panicle initiation, flowering and grain filling—resulting in a total of four unique predictor variables (one per stage) for each meteorological and agronomic predictor.

For each temporal model specification, predictor variables were aggregated by either averaging daily values within the period (for minimum and maximum temperature) or by calculating the cumulative sum (for precipitation) or integration (for LAI) of the variables over the specified period. Each of the 28 models was fit using random forests (RF) [32], a cumulative learning algorithm for regression and classification problems based on decision trees and bagging (bootstrap aggregation). Many studies have demonstrated the effectiveness of the RF model in modelling agricultural biophysical processes, particularly those that are nonlinear [33–35]. During the training process, RF builds a 'forest' from regression trees that are developed from a bootstrap sample of input datasets. Each bootstrap sample contains two-thirds of the input dataset, whilst the remaining samples that

are not included in the training are used to validate the model and assess the importance of predictor variables. Once the model construction terminates, predictions can be made by considering the expected value of all individual predictions of regression trees in the forest. We performed this RF analysis using the randomForest package v 4.6-14 [36] in R, and considering the default parameters (e.g., number of trees) suggested by package developers in R environment [37].

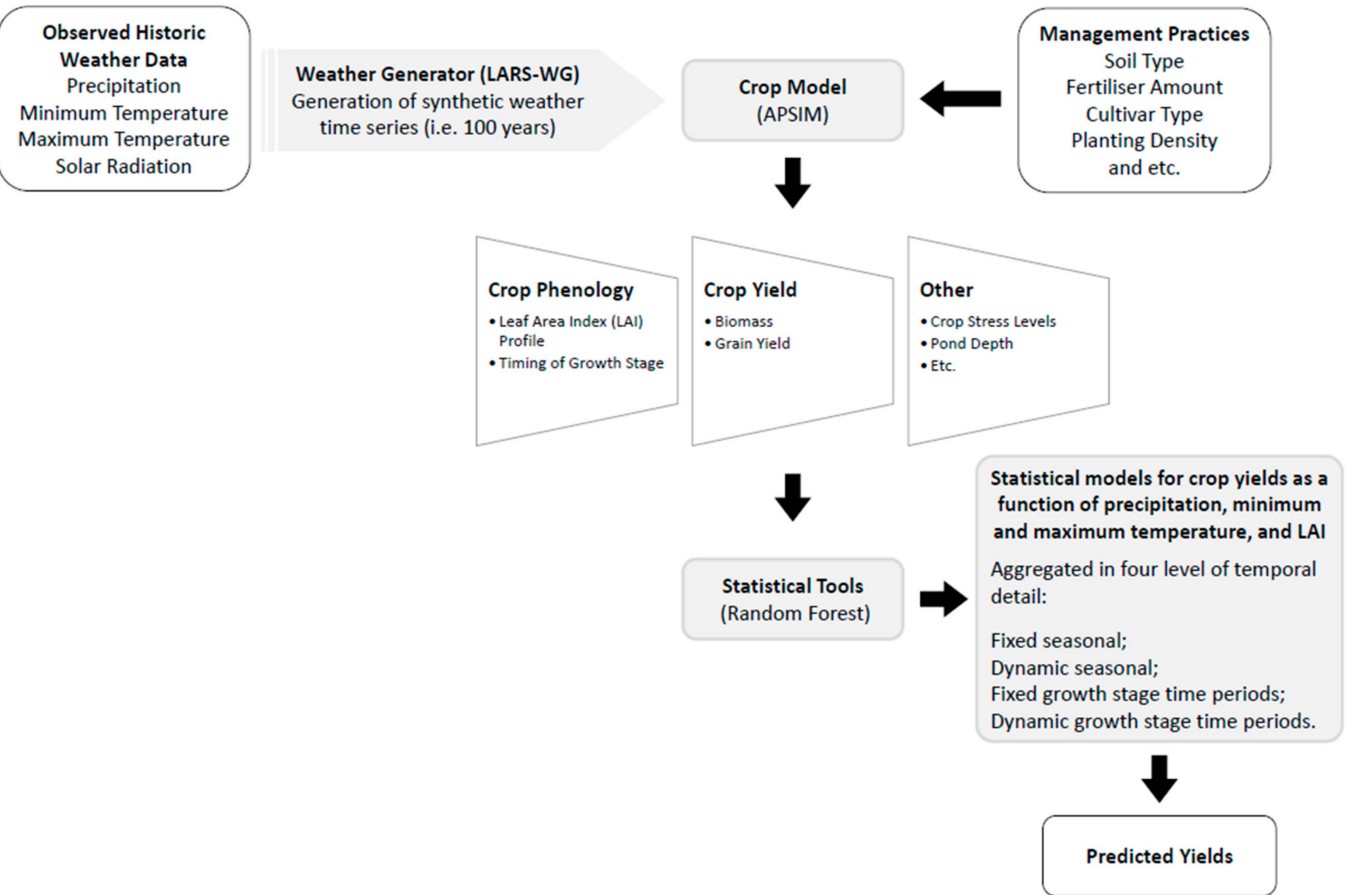

**Figure 2.** Overall procedure of coupling process-based crop models and statistical models to emulate synthetically generated Agricultural Production System sIMulator (APSIM) crop yields. First, APSIM was used to simulate crop phenology, LAI, yields and other state variables for a range of potential weather conditions and management practices. Through random forest analysis, statistical index relationships were then developed relating simulated yields to different combinations of weather and agronomic predictors varying in their combination and temporal aggregation.

### 2.4. Validation of Statistical Yield Models

We first compared the ability of each alternative model design to reproduce synthetic yields simulated by APSIM, focusing on the R-Squared and Root Mean Square Error (RMSE) of yield estimates in comparison with the actual APSIM simulated yields and the rates at which an area yield index-based insurance policy would produce false positives (upside basis events) and false negatives (downside basis events). To generate these measures of basis risk, we considered a policy that triggers a payout when predicted yields fall below 70% of the potential yield (i.e., 70% of the maximum observed crop yield for the same management practices) [38]. For all of these analyses (including false positive and false negative ratios analysis), we initially split 201,600 simulated yield observations into training and validation samples through a random selection, considering 80% of the observations as training data and the remaining 20% of the observations as data not used during the development phase of the statistical models.

In the simulations, noise was introduced by the parameters that enter the biophysical crop model, but empirically, these were not the only source of noise. When analyzing model performance in terms of predicting empirical yields, one should also be concerned about measurement error. To analyse whether introducing random noise affects the relative performance of the different RF models, after fitting different RF models to the training samples, we applied a mean-preserving spread to yields in the validation sample. Specifically, we added a normally distributed noise term to normalised yields in the validation sample whilst varying the standard deviation of this noise term between zero (no noise) and one (noise value with variation equal to the variation of observed yields in the validation sample). We then graphed the performance of our statistical yield models against the standard deviation of this random variable.

After identifying the best performing statistical model for emulating APSIM simulated yields, we sought to evaluate the ability of this model to reproduce observed yields in our study area. To determine yields for each of the 80 unique fields in our observed yield dataset (Section 2.1), we obtained weather and crop development observations for the 2019 rice-growing season over our study region. Daily time series of precipitation and temperature (minimum and maximum) were obtained from the ERA5 reanalysis dataset at $0.25° \times 0.25°$ resolution [39]. Timing of rice growth stages was determined from normalized difference vegetation index (NDVI) time series–interpolated from discrete values obtained from Sentinel-2 satellite imagery (with a spatial resolution of 10 m)–for each of the 80 fields (with an average plot area of 630 square meters; ~approximately equal to the area of six Sentinel-2 pixels) in our sample. Specifically, we assumed that the minimum NDVI value at the inflection point on the rising limb of the curve corresponded with the transplanting date and that harvest occurred when the NDVI time series equaled 0.25 on the falling limb in line with standard practices [40–42]. Timing of other growth stage transitions was determined based on the typical time from transplanting to reach the start of each stage for Swarna rice: 41 days, 61 days and 75 days for the start of panicle initiation, flowering and grain filling, respectively.

LAI time series for each field were generated based on spectral band data obtained from Sentinel-2 and Landsat-8 imagery retrieved through Google Earth Engine [43]. Estimates of LAI were generated for each cloud-free pixel based on spectral band data provided by Sentinel-2 and Landsat-8 imagery using an inverted radiative transfer model (RTM) [44]. The RTM inversions were developed by running the PROSAIL model 5000 times to generate synthetic reflectance data for a range of possible combinations of rice canopy, leaf and soil properties. The canopy, leaf and soil parameters used within the PROSAIL RTM simulations were drawn from a truncated normal distribution (TND) so that their values followed a normal distribution and not fall below zero at the same time (except for the solar zenith angle and relative azimuth angle; Table 1) following parameter ranges reported in previous applications for rice LAI estimation [45,46]. We used the simulated reflectance data to develop statistical models between the spectral bands collected by Sentinel-2 and Landsat-8 to LAI using a procedure similar to the RF approach described previously in Section 2.3. To develop statistical models for LAI, we split data equally at random between training and validation. The validated RTM model was subsequently used to convert observed reflectance on a given field into a discrete estimate of rice LAI for each available cloud-free observation, which was then converted to a continuous LAI time series for each field by fitting a double logistic function.

Estimated yields were compared with observed data from CCEs using R2, RMSE, and Normalised Room Mean Square Error (NRMSE) statistics at two spatial scales: (1) plot scale (80 observations with an average area of 630 square meters), and (2) GP scale (20 observations, with an average of 4 plots per GP). As noted previously, the latter equated to a spatial scale similar to a cluster of nearby villages (with an average area of 48 square kilometers ranging between 0.2 and 102 square kilometers), which is the lowest level of governing institutions in India's administrative structure. Importantly, GPs form the primarily spatial unit for area-yield insurance within the Indian government's national

crop insurance program, which at present relies on data from manual CCEs to verify crop yield losses and any resulting payouts to farmers. Understanding the performance of our methods at this scale is, therefore, of particular importance for understanding potential opportunities and challenges for satellite data and crop models to help reduce costs and time associated with crop insurance in India.

**Table 1.** Canopy, leaf and soil parameter ranges and distributions used within the PROSAIL radiative transfer model (RTM) simulations.

| Parameter Name | TND Mean | TND Std. | TND Lower Bound | TND Upper Bound |
| --- | --- | --- | --- | --- |
| Structure Parameter | 1.5 | 0.3 | 1.2 | 2.2 |
| Chlorophyll content (ug/cm$^2$) | 45 | 30 | 20 | 90 |
| Brown pigment content | 0.25 | 0.1 | 0.1 | 0.5 |
| Dry matter content (g/cm$^2$) | 0.05 | 0.005 | 0.003 | 0.011 |
| Dry/Wet soil factor | 0.9 | 0.25 | 0..3 | 1.2 |
| Leaf area index (m$^2$/m$^2$) | 3.5 | 4.5 | 0 | 10 |
| Leaf angle distribution | 60 | 20 | 30 | 80 |
| Hotspot parameter | 0.2 | 0.2 | 0.1 | 0.5 |
| Parameter Name | Uniform Distribution (minimum value) | | Uniform Distribution (maximum value) | |
| Solar zenith angle (deg) | 15 | | 90 | |
| Observer zenith angle (deg) | −12 | | 12 | |
| Relative azimuth angle (deg) | 6 | | 6 | |

## 3. Results

### 3.1. Performance of Statistical Models in Emulating APSIM Simulations

We first compared the performance of different model specifications (with varying predictor variables and varying levels of temporal aggregation for these variables) in terms of their ability to reproduce the synthetic yield data generated using APSIM and were perturbated with different noise levels. This is equivalent to selecting the best performing index for the design of an index insurance product. Figure 3 summarises the performance (left: R$^2$, right: RMSE) of each candidate model at different noise levels (with the standard deviation of the random noise term varying from zero to one, to represent cases without noise and cases with high degrees of noise, respectively) in predicting the validation data not used in the training phase of the RF models. Later, using an illustrative, synthetic insurance policy, we evaluated the basis risk of different model specifications by comparing across models the rates at which this synthetic policy would produce false positives (downside basis event) and false negatives (upside basis event) if triggered using yields estimated using our different statistical crop yield model specifications (Figure 4).

From inspection of Figures 3 and 4, several findings emerge. As expected, the model accuracies in terms of R$^2$ and RMSE statistics, as well as false positive and false negative frequencies for all model combinations and temporal disaggregation levels, decreased as we introduced more noise to yields in the validation datasets (in addition to the noise in APSIM simulated crop yields stemming from different weather realisations and management practices). However, the relative performance of models according to their error statistics used in this study (R$^2$ and RMSE as well as false positive and false negative frequencies) did not change by adding random noise to the validation datasets. This implies the efficiency of our validation efforts in picking the most suitable statistical model for crop yield estimation.

Moreover, the models more accurately reproduced the heterogeneity in APSIM-simulated rice yields when considering greater temporal disaggregation of predictor variables (i.e., moving from top to bottom in both columns of Figure 3). The greatest improvement in model performance–observed for all standard deviations considered when introducing random noise—was found when including predictor variables disaggregated by crop growth stage (DSTG models). Across potential models, the RMSE error reduced by 4% to 46% (with an average of 24%) and 2% to 24% (with an average of 11%) when input



variables were aggregated by fixed and dynamic growth stage (as opposed to the fixed and dynamic seasonal, FS and DS, models), respectively. Moreover, the reduction in RMSE from aggregating predictor variables using dynamic instead of fixed growth stages reduced RMSE values by 3% to 52%, with an average reduction of 24% across models considering different combinations of temperature, precipitation and leaf area index predictors.

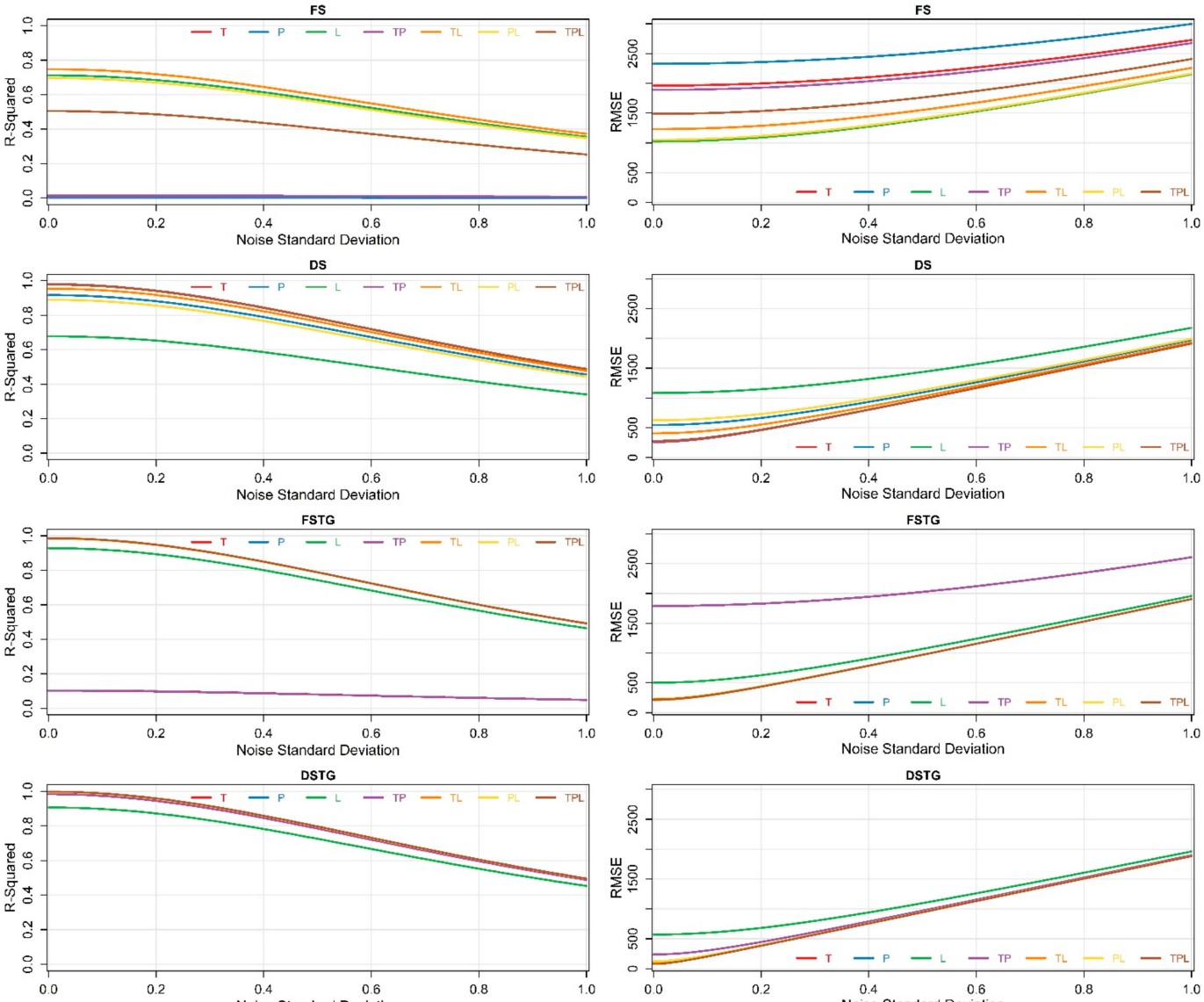

**Figure 3.** Performance of statistical models in emulating synthetic APSIM point-scale simulated rice yields at different noise levels (horizontal axis). Panels are labelled with the temporal aggregation of predictors, and colours of line plots in each panel denote the different model predictor variables considered in each model specification. The vertical axis shows either $R^2$ (left) or RMSE (right) for out-of-sample validation based on a retained 20% of the 201,600 unique APSIM yield simulations perturbated by different noise levels. T: Temperature, P: Precipitation, L: LAI, FS: Fixed Season, DS: Dynamic Season, FSTG: Fixed Crop Growth Stage, DSTG: Dynamic Crop Growth Stage.

Another noticeable trend from examining Figure 3 is that model performance improved by integrating multiple weather and crop predictor variables, regardless of the magnitude of the mean-preserving spread that we applied by introducing random noise to yields in the validation sample. A comparison of alternative model configurations using unseen validation datasets in Figure 3 (i.e., comparing seven continuous lines shown with different colours) shows that the best model performance—in terms of both $R^2$ and RMSE

at different noise levels—was achieved when combining temperature, precipitation and leaf area index predictors. Reliance on a single variable alone appeared to reduce the capacity of our models to capture the variability in crop yields simulated by APSIM accurately. Leaf area index alone was found to be the least robust individual predictor of yields, with models based on temperature, precipitation or a combination of these two variables generating significantly more accurate yield estimates. For example, for dynamic stage model specifications, on average (i.e., across all of the considered noise levels), the RMSE of models considering only leaf area index as a predictor was 1147 kg/ha compared with 993 kg/ha for models including only weather predictors (temperature and precipitation), an increase in RMSE of approximately 15% when only using leaf area index as a predictor of yields.

In the context of agricultural index insurance, it is important that methods of yield estimation minimise risks to farmers and insurers associated with either false negative or false positive events. Considering a hypothetical insurance contract where payouts are triggered if estimated yields fall below 70% of yield potential (defined as the maximum yield across perturbated simulated weather scenarios and management practices), we found that performance of different model specifications in terms of false negative and false positive events showed broadly comparable trends that were visible over $R^2$ and RMSE statistics (Figure 4). A decreasing trend of false negative and false positive frequencies was observed when moving from FS to DSTG aggregation of yield predictors (moving from left to right columns in Figure 4).

Similarly, reductions in false positive and negative were also found for statistical models that considered both weather and LAI information simultaneously (TPL model configuration variation shown with brown colour; Figure 4) as opposed to model types that used only weather variables as predictors (T, P and TP model configurations shown with red, blue and purple colours, respectively; Figure 4). Overall, we found that the total basis risk (average of summation of false positive and false negative risks across all of the considered noise levels) associated with conventional contracts (i.e., fixed season or fixed stage weather index contracts) could be potentially decreased by 16% by implementing yield estimation methods that utilised phenology specific information on both meteorological and crop growth conditions.

This reduction in basis risk diminishes, however, as we introduced more noise to the reference datasets during validation. It is important to keep in mind that measured basis risk in insurance contracts will be higher—and basis risk reductions more difficult to realise—when analyzing model performance using empirically observed yields, which will inevitably include more noise than the simulated yields in our validation datasets.

### 3.2. Performance of Statistical Models in Estimating Crop Yields at Plot and GP Levels

For application in the context of agricultural insurance, it is important to assess the ability of statistical models to reproduce not only synthetic yields simulated by APSIM but also observed yield in real-world smallholder farming environments. We, therefore, evaluated the ability of our best performing 'index' (i.e., model specification considering temperature, precipitation and LAI predictors disaggregated by growth stage; see Sections 2.4 and 3.1) against observed yields from CCEs conducted in the Jajpur district in the state of Odisha in eastern India, at the plot level (80 observation points) and averaged by GP level (20 observation points). It is noteworthy that we did not aim to identify the best-performing model using the observed yield data because of limitations in observed yield data availability. Specifically, a lack of sufficient variation across space and time to reliably estimate our statistical yield models, whilst this is the exact same challenge in index design that motivated our use of crop simulation models.

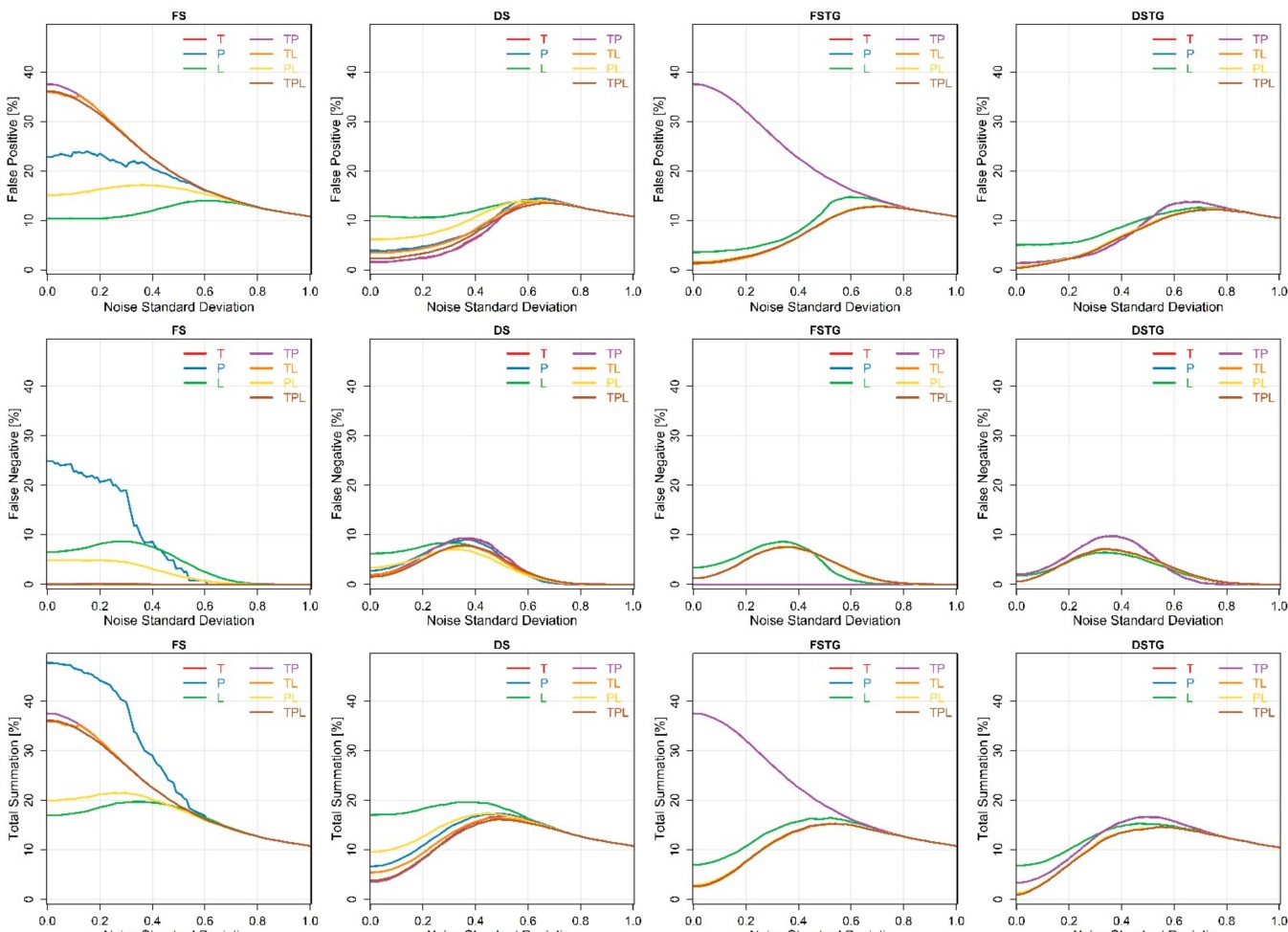

**Figure 4.** The frequency of basis risk in terms of false positive (%), false negative (%), and total summation (%) of them (shown in top, centre and bottom rows, respectively) associated with different model combinations (shown by different colours), heterogeneity levels (shown in different panels differing by columns), and introduced noise levels (horizontal axis) considered in this study. T: Temperature, P: Precipitation, L: LAI, FS: Fixed Season, DS: Dynamic Season, FSTG: Fixed Crop Growth Stage, DSTG: Dynamic Crop Growth Stage.

We find that our statistical model, developed based on the synthetic yield data simulated using APSIM, explained around 54% of the variance in observed rice yields at GP level, with an RMSE of 546.27 kg/ha (Figure 5b and Table 2). Performance of yield estimation at plot level was lower, with our model able to explain approximately 26% of observed yield variability with an RMSE of 860.1 kg/ha (Figure 5a and Table 2). Moreover, the analysis of false positive and false negative ratios of the estimated crop yields demonstrates that our model could estimate crop yields with 22.5% and 20% of false positive ratios and 12.5% and 0% false negative ratios at plot and GP levels, respectively. Model accuracy was lower when predicting observed yields than when predicting APSIM simulated yields, likely due to constraints imposed by the spatial resolution of weather data, gaps in LAI time series caused by cloud cover, and uncertainties in the underlying radiative transfer model used to translate spectral data obtained from Sentinel-2 and Landsat-8 into estimates of rice LAI during the season. However, an $R^2$ of 0.54 for GP-level yields suggests that such an approach may offer a useful tool for the design of index insurance products at this scale, for example, in the context of supporting area-yield index insurance products within the Indian Government's PMFBY crop insurance program.

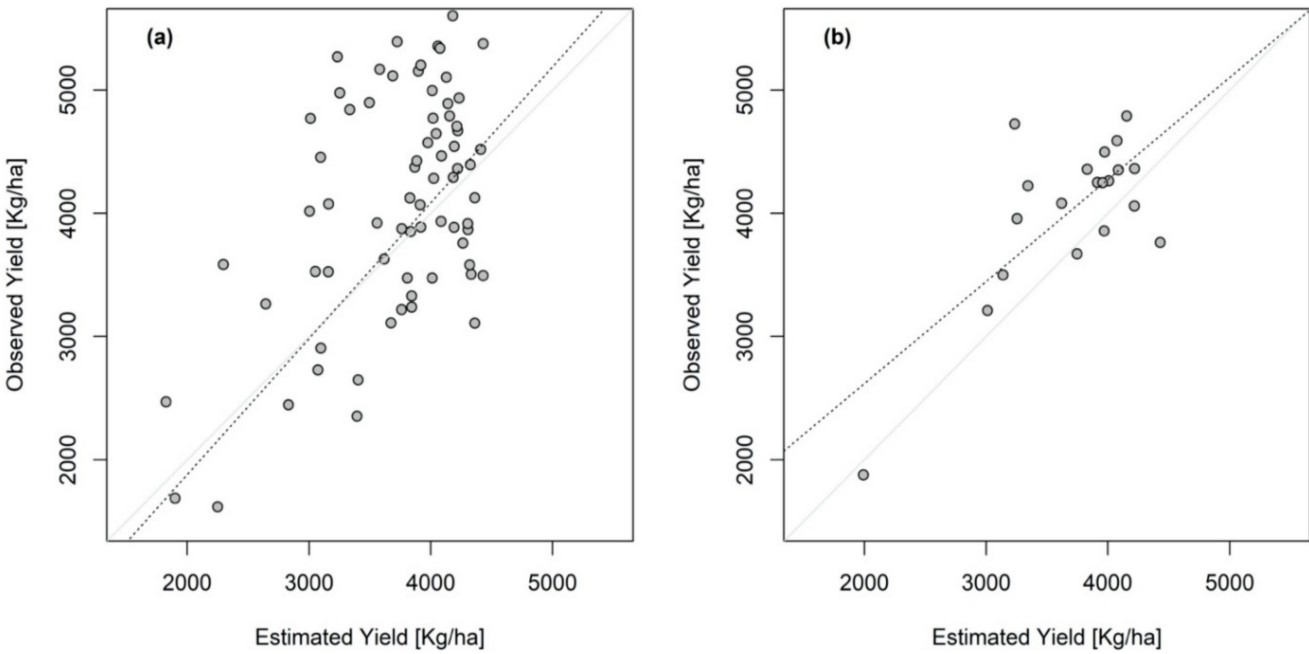

**Figure 5.** Scatterplot comparing estimated and observed rice yields (kg/ha) at (**a**) plot, and (**b**) Gram Panchayat (GP) spatial scales.

*3.3. Comparison of Statistical Models Developed Based on Vegetation Indices and Process-Based Crop Model Simulations*

As previously highlighted, a common challenge when developing and designing index insurance products is the limited availability of yield observation datasets to train underlying models that quantify yield losses based on proxy data. To evaluate the value provided by using crop models to generate larger synthetic yield training datasets, we compared the performance of our models reported in Section 3.2 with plot and GP level yield estimates derived using statistical vegetation index (VI)-based models developed by using observed yield data from CCEs. This is a common approach underlying the design of many existing index insurance products in India and other parts of the world [47–49], and, as such, understanding what, if any, improvements in accuracy are obtained from using crop models alongside satellite data is critical to understand the value-added from more complex approaches.

We considered two alternative potential VI-based yield models, using as predictors the average values of either NDVI or EVI. In each case, the used predictor was disaggregated by averaging over the four main rice crop growth stages to match the best performing model stemming from Section 3.2. We developed each statistical VI model using the RF approach, with VI values for each field and growth stage calculated using spectral bands data and observations of growth stage timings obtained from Sentinel-2 and Landsat-8 for each field, as described in Section 2.3. Note that we trained these alternative models by using a bootstrap approach (1000 times) and validated them through a k-fold validation scheme by considering 5-folds with 16 observations in each of them to minimise risks of overfitting, which could occur if all yield data were used in training and validation. We did not include weather data as an additional predictor in these models, given that the coarse resolution of available gridded weather products means that there was an insufficient spatial variation to adequately capture the effects of spatial weather heterogeneity in model training. Moreover, it is common for index insurance products to typically consider only vegetation indices or only weather data as predictors, with relatively few attempts to date to combine these predictors in index insurance contracts.

Similar to the trends observed in the previous analysis of statistical yield models that were derived based on APSIM simulated yield data, we found an improvement in the per-

formance of VI-based models when aggregating yield estimation from plot to the GP level (Table 2). However, comparing the performance of VI and crop model derived yield estimates at the same spatial scale, predictions by our crop model approach far outperformed the predictions from both VI models. For example, at GP scales, our preferred model captured 54% of observed yield variability, whilst VI-based models captured on average only 39% of yield variability (36% and 42% for the NDVI- and EVI-based model, respectively). VI-based models are also associated with larger RMSEs; on average, RMSEs were 12% higher than in the statistical yield model that we estimated from APSIM simulations.

**Table 2.** Accuracy assessment of statistical models developed based on vegetation indices and process-based crop model simulations. CCE: Crop Cutting Experiments, GP: Gram Panchayat, Std: Standard Deviation.

| Model | Plot Level | | | GP Level | | |
|---|---|---|---|---|---|---|
| | $R^2$ (Std.) | RMSE (Std.) | NRMSE (Std.) | $R^2$ (Std.) | RMSE (Std.) | NRMSE (Std.) |
| NDVI | 0.11 (0.03) | 990.82 (36.09) | 0.24 (0.01) | 0.36 (0.07) | 633.45 (47.05) | 0.16 (0.01) |
| EVI | 0.11 (0.03) | 1000.20 (35.44) | 0.24 (0.01) | 0.42 (0.09) | 590.49 (47.63) | 0.15 (0.01) |
| Crop Model | 0.26 (0) | 860.08 (0) | 0.21 (0) | 0.54 (0) | 546.27 (0) | 0.14 (0) |

Moreover, VI-based models are also associated with higher levels of uncertainty depending on which data are included in model training (for instance, we found a 47.63 standard deviation of RMSE values of 1000 bootstrap EVI based yield estimations at the GP scale). Where the total number of yield observation data is limited, as is the case here, and is common in almost all smallholder environments, this result suggested that the use of APSIM or other crop models can play an important role in improving the accuracy and robustness of yield-index relationships necessary for designing index insurance products relative to satellite vegetation indices alone.

## 4. Discussion

Relative to other forms of insurance and risk financing, index insurance schemes can provide a relatively low-cost and easy-to-implement solution to protect smallholder farmers against production risks posed by extreme weather events and climate change [50,51]. However, the value of these products for farmers and insurers is strongly predicated on the ability to base insurance payouts on index relationships that reliably and accurately quantify crop yield losses at disaggregated spatial and temporal scales [10].

We show that combining crop modelling and satellite-based crop phenology measurements can provide a scalable solution for deriving the relationship between yields and proxy indices at spatial scales relevant for agricultural insurance. Our findings highlight that accounting for field-level heterogeneity in crop phenology and combining multiple types of predictor variables, including both weather and leaf area indices, can significantly enhance model accuracy, in particular when aggregating to spatial units larger than an individual plot; which is common for area-yield index insurance in smallholder farming systems, such as those in our study area [52]. It is also important to note in this regard that in the crop model simulations, we did not use the information of conducted CCEs (the CCEs information were used as an empirical validation), and we only used publicly available information on typical ranges of cropping practices and varieties to generate synthetic yield training data; we did not measure these variables to implement crop models at the plot level, which adds to the scalability of this approach. Our focus on rice in Odisha is a specific use case, but one could apply this approach in any other region, and for any other crop, by using a biophysical crop simulation model for the targeted crop, training a statistical yield model using the data arising from these simulations, and using this model, combined with actual weather data and data from remote sensing imagery on growth stages and LAI, to predict yields empirically in that region.

Our findings do not support a finding from previous research combining earth observation data and crop models: that yields can be estimated using either a single (peak

or aggregated total) value of LAI for the season or multiple LAI predictors that relate to specific satellite image dates but are not directly correlated with crop growth stage timings [53–55]. Due to the lack of variation across space and time of our field level observations, we were unable to validate all model combinations and aggregation levels with field measurements. Nonetheless, the comparison of models derived from simulated data in terms of their ability to capture heterogeneity in simulated yields suggests that disaggregating predictor variables by crop development stages enhanced the accuracy of predicted yields relative to simpler seasonal aggregation. This will be true, especially when heterogeneity in phenology between fields and seasons is large due to differences in farm management practices, crop varietal choices, and weather conditions. We also found that insurance index performance can be improved further by combining LAI and weather predictor variables, which we attribute to the ability of weather data (in particular temperature predictors) to capture crop yield losses associated with deficient grain filling or pollination that would not be fully captured by changes in LAI alone [56].

Although the value of phenology data for improving yield estimation and index insurance has been demonstrated previously [21,57,58], these studies were focused on developed countries where extensive and longstanding phenological monitoring networks exist. We show that it is possible to replicate some of these improvements in yield estimation accuracy, and we highlight for smallholder environments the potential to reduce basis risk in index insurance using satellite-derived information on the timing of key development phases. The value of phenological information was largest when considering not only heterogeneity in the timing of the start and onset of the crop growing season but also in the timing of specific individual growth stages. This result is consistent with evidence suggesting that the effects of extreme weather on yields of rice and other crops are strongly dependent on the timing of shocks during the season, with the potential for larger yield losses if weather-related shocks occur during critical growth periods, such as anthesis [25,59]. Critically, only adjusting the seasonal time period for index insurance contracts, for example, to account for potential impacts of delayed transplanting of rice in years for with late monsoon onset [24], would fail to exploit the true value of phenological information for yield estimation.

Whilst our results suggest potential benefits of using crop model simulations to support the design of agricultural index insurance products, several approaches could be used to improve the accuracy of yield estimation at the plot level, which would aid both the design of plot-level index insurance and the accuracy of larger-scale area-yield index insurance. For example, in this study, we relied on a relatively simple satellite-based method for estimating intra-seasonal crop phenology. Integration of in-situ imagery, for example, taken by farmers through smartphones at regular intervals during the season [22], could help to reduce uncertainties in satellite-derived growth stage timings whilst also providing a supplementary source of information to help to validate fitted LAI time series. Such data would be especially valuable for crops grown during the rainy season, a period where substantial gaps in satellite imagery often occur due to high levels of cloud cover. In addition, in-situ imagery could provide a mechanism for detecting crop damage that may be difficult to reliably correlate with weather or vegetation indices, for example, mechanical damage to crops caused by flooding, wind and hailstorms or pests and diseases [60]. These factors are a potentially important driver of errors in plot-level yield estimation, suggesting that integration of in-situ imagery should contribute to reducing basis risk, especially at these finer spatial scales.

A further factor that may explain the larger errors in yield estimations observed in our analysis at the plot versus GP scales is the coarse resolution of weather data available in our study region. The ERA-5 reanalysis dataset used in this study has a spatial resolution of $0.25 \times 0.25$ degrees (approximately 25 km $\times$ 25 km), which is sufficient to capture heterogeneity in weather conditions between GPs but not between individual plots within a GP. Given the important role of weather data in yield estimation (Section 3.1), this suggests that the provision of finer resolution weather data could play an important role in

supporting reductions in basis risk of index insurance products. However, the development and validation of fine-scale weather data products remain challenging in many smallholder environments due to the limited density and completeness of in-situ weather records [61], in contrast to more extensive monitoring networks found in regions such as Europe and North America [8].

Finally, a key finding from our analysis is that the use of crop models provides added value for yield estimation beyond the use of statistical models based solely on satellite vegetation indices. Nevertheless, it is important to note that whilst our analysis considered two of the most commonly used vegetation indices for index insurance and yield estimation (NDVI and EVI), alternative types and combinations could have been used. For example, studies by Enenkel et al. (2018) [12] and Mollmann et al. (2020) [62] showed that developing more complex statistical models using multiple types of vegetation indices from different satellite datasets (e.g., Sentinel-1 or Sentinel-2) can yield more robust crop yield information. Hence, future research should seek to evaluate a broader range of vegetation index models to explore further the added value provided by the integration of crop models alongside satellite and other data sources. Moreover, future analyses should also consider how trade-offs between the two types of methods are affected by the amount and completeness of observational yield data and satellite imagery used to train statistical VI-based models. We hypothesise that the added value of crop models will be highest in environments where observational yield datasets are smaller, where satellite imagery is strongly affected by cloud cover, and where small plot sizes pose a challenge for remote sensing with currently available resolutions of satellite imagery; each of these are common characteristics of smallholder farming environments that are the focus of this study.

## 5. Conclusions

Index-based insurance provides a potential solution to transfer risks caused by crop failure away from smallholder farmers, providing farmers with a timely payout in the event of a poor harvest without the need for expensive manual verification of yields as in the case of traditional indemnity insurance. However, basis risk, that is, a poor correlation between actual yield losses and losses estimated based on the insurance index, remains a key challenge to scaling index insurance, reducing farmers' willingness to pay for insurance products and their ability to adapt to climate variability and change. In this study, we evaluated the potential to improve the accuracy of index insurance by combining process-based crop models, satellite-derived phenological metrics, and geospatial weather data to design index insurance products, focusing on a case study of rainfed rice production in the state of Odisha in eastern India.

We showed that when accounting for field-level heterogeneity in crop development and timing of extreme weather events, it is possible to reliably estimate rice yields without the need for extensive observational yield training datasets and without having to apply real-time data-demanding plot-level crop simulations. Our analysis demonstrated that yield estimation is improved by considering both agronomic (i.e., leaf area index) and meteorological (i.e., temperature and precipitation) drivers of yield variability. Performance also increased when aggregating individual plot-level estimates to the village or GP-level scales, suggesting that approaches proposed in this paper may have value in reducing reliance on the time and resource-intensive CCEs that are typically used to support the assessment of losses in area-yield index insurance products in India.

Our findings further showed that the accuracy of yield estimation by our preferred crop model and satellite information approach significantly outperformed models based solely on satellite vegetation indices and was consistent with existing research using crop models and satellite data for yield estimation in India, even though these studies have typically focused on crops such as wheat where satellite imagery is much less affected by cloud cover. Overall, our results highlighted the potential of technologies, such as crop modelling and satellite remote sensing, to support smart phenology-driven index insurance contracts, with potential for further improvements in yield estimation accuracy

as high-resolution satellite and in-situ crop monitoring becomes increasingly viable in smallholder environments.

**Author Contributions:** The authors confirm the following contributions to the paper: study conception (M.H.A., T.F., B.K., K.H., S.M.), development of methods and collection of data (M.H.A., T.F., T.P.H., B.P., B.K., K.H., S.M.), analysis and interpretation of results (M.H.A., T.F., B.K., K.H., T.P.H.), draft manuscript preparation (all authors). All authors have read and agreed to the published version of the manuscript.

**Funding:** Funding support for this study was provided by NERC-ESRC-DFID Award No. NE/R014094/1; the Government of India, Ministry of Agriculture and Farmers Welfare, Department of Agriculture Cooperation & Farmers Welfare, Mahalanobis National Crop Forecast Centre; the CGIAR Platform for Big Data in Agriculture; and the CGIAR research program on Policies, Institutions and Markets (PIM), led by the International Food Policy Research Institute (IFPRI). This research was undertaken as part of PIM and the CGIAR research program on Climate Change, Agriculture and Food Security (CCAFS).

**Data Availability Statement:** The climate data used in this study are openly available in The ERA5 global reanalysis at doi:10.1002/qj.3803 (access on 18 February 2021).

**Acknowledgments:** We gratefully acknowledge Tharakeswar Ganta, Senthil Kumar and Shashank Bhushan Dash from Dvara E-Registry for their dedication to the ground data collection. Ben Parkes is the recipient of the Ekpe Research Impact Fellowship in the Department of Mechanical, Aerospace and Civil Engineering at the University of Manchester.

**Conflicts of Interest:** The authors declare no conflict of interest.

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
