# Peer review of "Improving the Performance of Index Insurance Using Crop Models and Phenological Monitoring"

_remotesensing, doi:10.3390/rs13050924_

Round 1

Reviewer 1 Report

Referee Report on “Improving performance of index insurance using crop models and phenological monitoring” (Manuscript ID: remotesensing-1061250) This paper tries to identify better crop yield predictors with the purpose to improve the performance of index insurance.

I find the paper interesting and well-written. I don’t have many comments. My major suggestion is to conduct some simulation to quantify the improvement of a typical index insurance to a representative farmer based on your preferred model when compared with a typical index insurance contract (e.g., indexed to growing season rainfall). Could you quantify how much basis risk might be reduced based on your preferred prediction model? Doing this will directly show the improvement on index insurance that your preferred model can generate.

Reviewer 2 Report

The research presented in the paper is very important and interesting.

The authors are advised to mention (in the Introduction) that there are also some other possibilities and approaches which could be used in future for crop insurance (apart from traditional insurance and index-based insurance). For instance, there are whole-farm revenue insurance :.

Chalise, L., Coble, K. H., Barnett, B. J., & Miller, J. C. (2017). Developing area-triggered whole-farm revenue insurance. Journal of Agricultural and Resource Economics, 27-44.

Ainollahi, M., Ghahremanzadeh, M. O. H. A. M. M. A. D., & Dashti, G. H. A. D. E. R. (2019). Evaluating the possibility of utilizing whole-farm revenue insurance in Zanjan city. Agricultural Economics (Karaj)13(2).

as well as bancassurance:

Dua, P., Sahay, D., & S Deol, D. O. (2019). An Overview and Significance of Different Bancassurance Schemes Launched for Financial Inclusion in India. International Journal of Management (IJM)10(6).

Chang, P. R., Peng, J. L., & Fan, C. K. (2011). A comparison of bancassurance and traditional insurer sales channels. The Geneva Papers on Risk and Insurance-Issues and Practice36(1), 76-93.

Reviewer 3 Report

Lines 174-177. Please comment on the spatial resolution of ERA5 v5.1.3 historical met data, related to the study area and observed yield data through CCEs. In other words, do all of your field data all into one ERA5 grid, or several? This may or may not be important for interpreting and discussing the results. 0.25 x 0.25 degrees comes up in line 253, but relating this to the observed yield data earlier on would help.

Section 2.2 Describe if these are point-based simulations, or gridded, and how they relate spatially to the observed data described in section 2.1

Line 255. Specify the spatial resolution of Sentinel-2, and compare it to the area of the 80 fields. Clarifying the temporal frequency of Sentinel-2 data would also help set up discussion on limitations (e.g., based on cloud cover, ability to capture peaks and troughs of NDVI related to the timing of growth stages)

Line 257. Please provide a reference for approximating harvest based on “NDVI time series equals 0.25 on the falling limb.”

Table 1. Please provide some justification on use of TND and the upper and lower bounds, either via references or justification specific to this study. It is not addressed in the cited Jacquemoud et al 2009 paper. Perhaps in M. Campos-Taberner et al. 2016. Multitemporal and multiresolution leaf area index retrieval for operational local rice crop monitoring. Remote Sensing of Environment; or L. Liang et al. 2020. Estimating Crop LAI using spectral feature extraction… Remote Sensing.  There is an extra decimal in the TND Lower Bound for Dry/Wet soil factor.  Add units where relevant to the first column.

Line 273. Similar comment to Line 255 on relating spatial resolution of Landsat-8 and Sentinel-2 to the field observation. You do not need to repeat this here, but an earlier comparison would help the reader understand the relationships.

Lines 282-286. This description is good and gets at the spatial resolution questions above. Please provide some indication or range of the area represented by 1) plot scale, and 2) GP scale.

Results / Figure 3. Specify the scale or aggregation of these results. Provide p values or confidence levels.

Line 305. Either a wording issue or missing point prior to this sentence—there is no “first notable trend” mentioned previously.

Line 336. Use of RMSE and R2 alone do not allow for an interpretation of false positives / false negatives, in the context of index insurance. The statement in lines 341-344 could be strengthened with additional interpretation of overall under/over prediction.

Line 514, and Discussion in general. I understand the motivation of finding alternatives to resource intensive CCEs, but I feel that a clearer argument could be made for the logical flow of the proposed improvement. The authors use comparisons between CCE and satellite-based statistically estimated yields to justify their next step in proposing process-based crop model as an alternative.  As the authors state, sufficient data are lacking to compare CCEs to APSIM simulated rice yields. I think this also means that they cannot quantify the “added value” of combining process-based crop models, satellite-derived phenological metrics, and geospatial weather data; versus CCEs alone. This paper provides a great framework for testing this hypothesis in other regions, but a question remains on scalability: to what extent are CCEs required to scale to other regions?

Round 2

Reviewer 1 Report

I don't have further comments. Good job!

Author Response

We sincerely thank the anonymous reviewer for the careful reading of our manuscript and for providing thoughtful comments that strengthened the quality and potential impacts of our manuscript.